# Microglial Phagocytosis—Rational but Challenging Therapeutic Target in Multiple Sclerosis

**DOI:** 10.3390/ijms21175960

**Published:** 2020-08-19

**Authors:** Maria V. Pinto, Adelaide Fernandes

**Affiliations:** 1Neuron-Glia Biology in Health and Disease, Research Institute for Medicines (iMed.ULisboa), Faculty of Pharmacy, Universidade de Lisboa, 1649-003 Lisboa, Portugal; mariagvp2009@gmail.com; 2Department of Biochemistry and Human Biology, Faculty of Pharmacy, Universidade de Lisboa, 1649-003 Lisboa, Portugal

**Keywords:** demyelinating lesions, microglia, myelin phagocytosis, multiple sclerosis, therapeutic strategies

## Abstract

Multiple sclerosis (MS) is the most common autoimmune and demyelinating disease of the central nervous system (CNS), characterized, in the majority of cases, by initial relapses that later evolve into progressive neurodegeneration, severely impacting patients’ motor and cognitive functions. Despite the availability of immunomodulatory therapies effective to reduce relapse rate and slow disease progression, they all failed to restore CNS myelin that is necessary for MS full recovery. Microglia are the primary inflammatory cells present in MS lesions, therefore strongly contributing to demyelination and lesion extension. Thus, many microglial-based therapeutic strategies have been focused on the suppression of microglial pro-inflammatory phenotype and neurodegenerative state to reduce disease severity. On the other hand, the contribution of myelin phagocytosis advocating the neuroprotective role of microglia in MS has been less explored. Indeed, despite the presence of functional oligodendrocyte precursor cells (OPCs), within lesioned areas, MS plaques fail to remyelinate as a result of the over-accumulation of myelin-toxic debris that must be cleared away by microglia. Dysregulation of this process has been associated with the impaired neuronal recovery and deficient remyelination. In line with this, here we provide a comprehensive review of microglial myelin phagocytosis and its involvement in MS development and repair. Alongside, we discuss the potential of phagocytic-mediated therapeutic approaches and encourage their modulation as a novel and rational approach to ameliorate MS-associated pathology.

## 1. Introduction

Multiple sclerosis (MS) is the primary chronic autoimmune, demyelinating disease of the central nervous system (CNS), and the most debilitating condition in young adults aged between 20 and 45 years [1]. It is mostly accepted, nowadays, that a dysregulation of the inflammatory response against myelin components-genetic and/or environmentally-mediated-leads to the recruitment of peripheral autoreactive T and B cells and macrophages across the blood-brain barrier (BBB). Reactivation of such immune cells within the CNS induces a potent inflammatory response with cytokine and chemokines’ release that further activates other immune cells (e.g., dendritic cells) and local microglia causing myelin to degenerate around the nerve axons. Myelin loss then leads to the subsequent formation of the MS-characteristic demyelinated plaques associated with axonal damage and neurodegeneration that clinically translates into initial relapses evolving with time into progressive disability of motor, visual, and cognitive functions, essentially [2,3,4,5,6].

MS prevalence and impact in life quality together with the significant financial burden either on the patient, society, and healthcare system potentiated the extensive demand and success of some therapeutic approaches using disease-modifying drugs approved because of their efficacy to reduce relapse rate and slow disease activity and progression [7,8]. Based on the above described autoimmune nature of MS, these immunomodulatory drugs either target T cell proliferation (interferon-beta) [9], immune cell activation and inflammatory response (interferon beta, glatiramer acetate) [10,11], or the peripheric invasion of autoreactive T and B cells: Preventing the ability of these cells to egress from the lymph nodes (fingolimod) [12]; or inhibiting their crossing over the BBB (natalizumab) [13,14]. Further revolutionary, cell-depleting drugs (e.g., alemtuzumab, ocrelizumab) [15,16] were also approved potentiating either the depletion of autoreactive lymphocytes or CD20-expressing B cells only, respectively, therefore modulating immune response through immune repopulation. Yet, they all failed to restore CNS myelin and to fully prevent MS disability.

Regarding MS recovery, current studies have been focused on remyelination, which is a biological regenerative process that restores myelin around denuded axons and has been documented in experimental models and acute MS lesions following inflammatory-mediated demyelination. Here, oligodendrocyte (OL) precursor cells (OPCs) are spontaneously recruited to the lesioned region, differentiating into myelinating OLs to restore myelin sheaths and protect them from further degeneration [17,18,19]. However, despite the presence of functional OPCs, the majority of lesioned areas with progressive demyelination in MS fail to remyelinate [20,21,22]. As an attempt to find remyelinating therapies only, clemastine fumarate validated its efficacy on remyelination in clinical trials by promoting OPC differentiation in chronic demyelinating lesions of MS [23], for which there is the need for other candidates to rescue MS lesions from progressive degeneration.

Microglia, the resident immune cells and macrophages of the CNS, make up only 15% of the total brain cells [24] and 10% of whole glial cells [25,26]. Nonetheless, due to their essential functions as immune mediators and primary phagocytes of the brain and spinal cord, microglia have attracted much attention and research. This yolk sac-derived population colonizes the brain during embryogenesis and differentiates under the influence of CNS microenvironment [27], residing in the healthy nervous system as a highly stable population with long and ramified processes interacting with blood vessels, neurons, and other glial cells in a dynamic “surveillant state” [28]. “Surveillant” microglia maintain CNS homeostasis by sensing pathologic events, scavenging pathogen-associated molecular patterns (PAMPs)/danger-associated molecular patterns (DAMPs), and by phagocytosing dead cells and misfolded proteins [29,30]. At the same time, in physiological conditions, they are also engaged in the regulation of biological processes either remodeling synaptic plasticity [31,32], supporting neuronal survival [33], brain sexual differentiation [34,35], or promoting proper myelination [36,37,38].

In response to any insult or alteration in brain homeostasis, these highly sensitive glial cells migrate and accumulate at the lesioned area, through a process of chemotaxis, and become activated to initiate the innate immune response. Profound morphological and molecular changes accompany microglial activation (Figure 1): Cells suffer partial retraction of their processes evolving from a ramified state, to an intermediate one or “Bushy” and finally converting into a complete rod and amoeboid shape or fully activated state [39,40]. Parallel to their morphological transformation, activated microglia express a characteristic genetic profile with an elevated expression of ionized calcium-binding adaptor molecule-1 (Iba-1), major histocompatibility complex antigen class II (MHC-II), and microglial phagocytic markers such as CD11b, Fc receptors I–III, complement receptors, and CD68 [41,42]. Alongside, functional plasticity allows these reactive/activated microglia to differentiate into either neurotoxic or neuroprotective phenotypes associated with the release of both pro- and anti-inflammatory mediators and cytokines, respectively. Given their wide range of functions, microglia participate in the overall brain development and formation of a functional nervous system and have also been extensively described under several pathological conditions [43,44,45]. Particularly, in MS lesions, microglia account for the main inflammatory cells present, therefore strongly influencing demyelination and plaque recovery. Alongside, the unbalance between opposed phenotypes of microglia (with overactivation of these cells into their pro-inflammatory status) accounts for disease development and progression, and correlate with the clinical decline, whereby we believe that the development of therapies targeting microglia will likely define another milestone in MS remyelinating therapeutics [46,47,48].

Indeed, emerged microglia-based therapeutic strategies have been mostly focused on the suppression of microglia-mediated inflammatory response and oxidative damage. Current research focused on anti-inflammatory molecules such as naringenin [49,50], ethyl pyruvate [51], and forskolin [52] envisioning regenerative approaches in MS by reducing microglial activation. Alongside, pre-clinical testing with a new vaccine, PADRE-K v1.3 [53], resulted in a decrease of pro-inflammatory microglia, in time replaced by an anti-inflammatory/regenerative population, alleviating clinical severity and pathological damages in a mice model of MS-experimental autoimmune encephalomyelitis (EAE). Furthermore, long noncoding RNA growth arrest-specific 5 (GAS5) that is highly expressed on amoeboid/activated microglia in MS lesions inhibits microglial polarization into a regenerative phenotype. Recent approaches on transplantation microglia with GAS5 interference ameliorated the EAE outcome, favoring as well axonal remyelination after LPC-induced demyelination [54]. Overall highlighting the experimental success of modulating the inflammatory role of microglia to rescue phenotypic unbalances (reviewed in [55,56]).

Nonetheless, remyelination failure also relies on the excessive accumulation of fragmented myelin-toxic debris in the extracellular space after prolonged demyelination, which significantly inhibits OPC differentiation. As so, apart from their crucial immune role in MS progression, clearance of myelin debris through microglial phagocytosis is required for tissue repair and critical in the progress of demyelinating diseases [57,58]. Whereas the microglial immunomodulatory role regarding MS has been extensively reviewed over the years, their contribution to MS disease progression as professional phagocytes, as well as therapeutic opportunities towards modulation of phagocytosis have been less explored. Therefore, in this review, we focus on the microglial phagocytic role throughout brain development and with a particular focus on MS-disease course, highlighting some molecules and targets with the potential to increase myelin clearance for disease amelioration but mostly for MS recovery.

## 2. Microglial Phagocytosis

Within the CNS, phagocytosis is mainly attributed to the specialized microglial cells, although other cells may slightly contribute (e.g., astrocytes, invading monocytes, and neutrophils). Microglial cells recognize, engulf, and digest large particles/structures, a process that is critical both in development, to remove apoptotic, excessive newborn cells, or unwanted synapses, and in pathology by clearing invading bacteria, cell debris, or degenerated myelin [59]. Indeed, microglial phagocytosis goes beyond brain refinement in physiologic conditions. It is critical in pathological conditions such as MS. Therefore, we will first address the microglial phagocytosis relevance during neurodevelopment and afterwards, the microglial phagocytic role in MS-like pathogenesis. Then, we will shed light on some promising phagocytose-based therapeutic strategies that could counteract MS disease burden, prevent demyelination, or even promote full disease recovery through remyelination.

### 2.1. Microglial Phagocytosis during Neurodevelopment

In the developing brain, non-activated microglia continuously survey the brain’s environment, making frequent but transient contacts with neurons and synapses in response to neuronal activity [60]. Neuronal expression of fractalkine (CX3CL1) as well as their release of ATP/ADP function as “find me” signals that will interact with microglial fractalkine (CX3CR1) and purinergic receptors (P2Y12), respectively, and guide them towards active synapses [61,62,63]. However, microglia not only rely on the vicinity of neurons and synapses. Instead, they are fundamental in the process of synaptic plasticity and proper refinement of neural networks through synaptic pruning: An activity-dependent developmental program in which microglia eliminate (through phagocytosis) supernumerary synaptic connections over CNS development [64,65]. Otherwise, deficiencies in synaptic pruning correlate with developmental abnormalities [66,67] and impaired social behavior [68].

Studies from Paolicelli et al., using stimulated emission depletion (STED) microscopy, illustrated the microglial direct role on the engulfment of pre- and post-synaptic components in a healthy brain [61], further confirmed in subsequent studies from Schafer et al. Using high-resolution confocal imaging, the authors found either microglial processes in close contact with pre-synaptic inputs of retinal ganglion cells or these synaptic partners internalized into microglial soma [65]. Interestingly, from in vivo studies, authors concluded microglial preferential removal of weaker synaptic inputs that were tagged by complement proteins C1q and C3, as deficient mice for either one of the latter complement proteins exhibited extensive defects in synaptic connectivity [65,69,70]. Yet, despite the involvement of complement cascade in neuronal elimination and pruning of synapses, other microglial receptors also play a role, such as CX3CR1 and triggering receptor expressed on myeloid cells 2 (Trem2) [71,72]. Particularly, Trem2 depletion on mice impaired not only cell migration towards hippocampal synapses but also synaptic internalization into CD68+ phagolysosomal structures. Furthermore, results from Trem2−/− old mice correlated with brain impaired functional connectivity between prefrontal and hippocampal regions and behavioral defects [72], as expected given the close association between phagocytosis and synaptic pruning with proper brain wiring, development, and function.

Additionally, the extensive cell turnover through apoptosis during brain development is again an important step for creating homeostasis and generating a functional brain. For that to be accomplished, cellular apoptosis drives microglial colonization and establishment in the CNS for a coordinated and efficient system of clearance/removal of apoptotic bodies by microglia, whereas inefficient phagocytosis of apoptotic bodies results, instead, in subsequent cell death alongside with inflammatory and immune responses because of the released intracellular contents [73,74]. Microglia are essential to remove apoptotic neural progenitor cells within the neurogenic niches of the hippocampus-subgranular and subventricular zones-via activation of TAM receptor tyrosine kinases MerTK and AXL [75], but also mediated by the complement system, as the C1q complement protein was recently found expressed on microglial phagocytic pouches during hippocampal neurogenesis [76]. Apart from the hippocampus, microglia also phagocytosed apoptotic Purkinje neurons, for the proper maturation of rat cerebellum and correct development of the retina in zebrafish, a process that is surprisingly delayed after inhibition of microglial purinergic receptor P2RY12, thus unveiling an additional role of purinergic signaling in developmental phagocytosis of apoptotic cells [74,77]. On the other hand, current studies on microglia associate cell phagocytosis with the brain’s sexual differentiation. Estradiol release in the female’s brain enhances the number of phagocytic microglia (observed by the observed augment of phagocytic cups), thus increasing the removal of neural progenitor cells in the neonatal female hippocampus. As expected, differences in the remaining neuronal progenitors resulted in developmental alterations, namely in the hippocampal neuronal wiring and excitatory circuits. Nelson et al. even speculated if such differences in phagocytosed products could account for functional disparities between males and females [34]. Alongside, phagocytosis is also involved in the sexual differentiation of the amygdala. Testosterone-mediated enhancement of 2-arachidonoylglycerol in the brain activates type-1 and -2 cannabinoid receptors in microglia that will increase cell phagocytic cups, resulting in the observed reduction of newborn astrocytes over the first postnatal week. By regulating astrocytic density, in a complement-dependent manner, microglia lead to an increased neuronal excitation and masculinization of juvenile social play, distinct from female behavior [35].

Altogether, these functional dynamics of microglial phagocytosis emphasize their importance in the overall brain development, maintaining functional homeostasis, controlling neuronal progenitors, and strengthening synaptic connections for proper brain connectivity and wiring.

### 2.2. Microglial Phagocytosis in Multiple Sclerosis: From Targets to Therapeutic Strategies

#### 2.2.1. Phagocytosis and Myelin Clearance

In MS, accumulation of chemokines, such as CXCL10, within demyelinated lesions function as chemoattractant molecules for microglia to migrate towards these regions where these cells will be responsible for the clearance of myelin-derived toxic debris. Nogo-A, oligodendrocyte-myelin glycoprotein, and myelin-associated glycoprotein are myelin-derived proteins and strong inhibitors of neurite growth, axonal regeneration, but most importantly, inhibitors of OPC differentiation into mature OLs and, by doing so, impair remyelination, which makes myelin clearance essential to initiate lesion repair [78,79,80]. The complement system is a well-described mechanism for the clearance of opsonized myelin as proteins of the complement bind to debris and control their interaction with complement receptor 3 (CR3) [81,82]. In addition to CR3, the microglial scavenger receptor AI/II (SRAI/II) can also recognize lipid debris and, when functioning together with CR3, even enhances complement-dependent myelin clearance by these cells [82]. Moreover, myelin also interacts with microglial MerTK, Trem2, and CX3CR1 receptors. Indeed, monocyte-derived macrophages from MS patients, with decreased expression of MerTK receptors, showed a reduced ability to clear myelin debris. On the other hand, Trem2−/− microglia not only lead to deficiencies in myelin removal but also in lipid metabolism, expression of inflammatory mediators, and trophic factors resulting in an augment of axonal dystrophy, oligodendrocyte reduction, and persistent demyelination in mice after cuprizone treatment [83]. Concordantly, mice treated with the anti-Trem2 antibody worsened EAE outcomes (high clinical scores), increased inflammatory response, and demyelination in the brain parenchyma [84]. Additionally, CX3CR1 deficiency in mice reduced significantly the clearance of myelin and inhibited proper remyelination after cuprizone treatment accentuating the association between myelin clearance and demyelination/degenerative mechanisms [58,85,86]. For this reason, skewing research approaches to modulate microglia towards a more phagocytic state offer promising alternatives envisioning a long-term MS treatment.

New compounds, in non-clinical testing, revealed some encouraging effects regarding microglial phagocytosis (Figure 2A). To begin with, studies from Chen et al. found that the incubation of primary microglia with two omega-3 polyunsaturated fatty acids-docosahexaenoic acid (DHA) and eicosapentaenoic acid (EPA)—modulated the cell phenotype towards a more regenerative state and enhanced myelin phagocytosis in vitro. Attractively, such treatment in vivo reduced demyelination and ameliorated motor and cognitive functions in a cuprizone demyelinating model [87]. Additionally encouraging, a human recombinant IgM antibody (rHIgM22) effectively modulated microglial myelin phagocytosis. Indeed, in vitro testing showed that rHIgM22 could bind to myelin, tagging debris for their internalization by microglia, in a complement-mediated manner [88], and, when administrated in vivo, rHIgM22 promoted OPC differentiation, accelerated remyelination, and even ameliorated memory deficits in rodent models of cuprizone and chronic virus-induced demyelination [89,90,91]. Giving this newly association between phagocytosis and microglial neuroprotective functions over brain pathologies, two more studies concerning modulation of microglial phagocytosis have just been published. In the first one, the authors treated rat microglial cells with endocannabinoid 2-AG. By enhancing the mRNA levels of phagocytosis associated genes (cd206, sirp1α, msr1, and Trem2), endocannabinoid 2-AG boosted phagocytosis of both Coli-coated beads and rat-purified myelin debris. Moreover, using a mice model of MS, the Theiler’s induced demyelinating disease model (TMEV-IDD), TMEV-induced mice treated with endocannabinoid 2-AG showed augmented microglial myelin phagocytosis in the corpus callosum that likely resulted from the observed upregulation of msr1 and Lamp1 mRNA levels, genes associated with microglial phagocytic machinery and phagosome formation, respectively. Interestingly, these animals had increased OPC differentiation and remyelination following demyelination in the corpus callosum [92]. In the second study, Liu et al. observed that a new pharmacological treatment with pseudoginsenoside-F11 accelerated CR3-dependent myelin phagocytosis in a microglial culture after oxygen-glucose deprivation (OGD) and permanent middle cerebral artery occlusion (pMCAO) in vivo and, by doing so, decreased de infarct area and improved neurological functions in pMCAO-treated rats [93].

These revolutionary approaches towards microglial-targeted therapies open novel windows of opportunity for MS remyelination assessment, for what other molecules should be taken into account in future MS-related experiments. It is described that the activation of peroxisome proliferator-activated receptor γ (PPARγ) in microglia suppresses the release of inflammatory mediators [94]. However, activation of PPARγ using both agonists: Pioglitazone and DSP-8658, also overexpresses the scavenger receptor CR36 and, while doing so, enhances microglial phagocytosis of Amyloid-β (Aβ) in primary microglial cell culture and in vivo using a mouse model of Alzheimer’s Disease (AD) [95]. Moreover, Yamanaka et al. reported a reduction of Aβ in the cortex and hippocampus, accompanied by improved spatial memory in treated mice, given this improved phagocytosis [96]. Another PPARγ agonist (ursolic acid (UA)), significantly decreased disease severity, CNS inflammation, demyelination, and promoted myelin repair through remyelination in both acute and chronic stages of the disease in EAE-induced and cuprizone-treated mice [97]. Whether the PPARγ-mediated effect of UA favor myelin phagocytosis by microglia among demyelinated lesions is unknown. However, we may postulate that once being upstream of the CD36 receptor that is also involved in the internalization of lipids and cholesterol-rich debris, PPARγ signaling may also interfere with cell phagocytosis and with the observed neuroprotective outcome on oligodendrocyte maturation and myelin regeneration following UA administration in vivo. On the other hand, treatment with nicotine or galantamine, this latter being an FDA approved acetylcholinesterase inhibitor for the symptomatic treatment of AD [98], significantly induced phagocytosis of extracellular toxic Aβ both in rat microglia and rodent models of AD. By sensitizing microglial nAChRs to choline and promoting calcium influx, galantamine and nicotine induced calcium-dependent massive actin reorganization favoring Aβ internalization, accompanied by improvements in spatial learning and memory performance. Therefore, one can speculate that galantamine-mediated phagocytic effects may also underlie its previously described benefits in AD [98,99], and may have an important role in myelin clearance following demyelination.

#### 2.2.2. Phagocytosis and Microglial Inflammatory Profile

The phagocytic process increases in complexity when the entrance of a large metabolic load from digested targets alters microglial intracellular pathways and metabolism. Particularly in MS, myelin by itself is a potent inflammatory stimulus so that its interaction with membrane receptors and further internalization can interfere with functional phenotypes of microglia and influence MS inflammatory responses, as reviewed by Grajchen et al. [100]. Whereas myelin entrance through the C3/CR3 axis initiates a subsequent inflammatory signaling cascade with the expression of neurotoxic cytokines and inflammatory modulators (via activation of the NF-kB signaling cascade) [101,102], Trem2-mediated myelin recognition, on the contrary, activates downstream anti-inflammatory pathways in response to demyelinating stimuli [83,84,103]. Subsequentially, over the course of myelin intracellular processing, activation of heterodimeric transcription factor, liver X receptor (LXR)-retinoid X receptor (RXR), and (PPARβ/δ) by myelin-derived cholesterol and phosphatidylserine residues, respectively, leads to downstream anti-inflammatory responses [104,105,106]. This being in line with the observed anti-inflammatory lipid-engorged (foamy) microglia within MS resolving lesions, preferentially expressing anti-inflammatory molecules and inducing T cell-mediated anti-inflammatory adaptative immune responses, even capable of reducing EAE severity [107,108].

Nonetheless, other studies describe the presence of pro-inflammatory and neurodegenerative foamy macrophages within MS lesions suggesting functional plasticity of these myelin-loaded cells [2]. Within foamy cells, myelin entrance is followed by lipid metabolization into myelin-derived free/unesterified cholesterol in the lysosomes. The toxicity of intracellular accumulation of free cholesterol requires their transportation, facilitated by lysosome membrane proteins such as Niemann-Pick type disease C (NPC) 1 and 2, to the endoplasmic reticulum for a process of esterification into cholesterol esters that will be posteriorly effluxed from the cell via ATP-binding cassette (ABC) transporters, or stored into lipid droplets [109], as depicted in Figure 2B. Despite the above correlation of myelin intracellular metabolization with an anti-inflammatory phenotype of microglia, it is observed that the uncontrolled entrance of cholesterol-rich lipids dysregulates degradation mechanisms as well as lipid storage and efflux pathways. Excessive intracellular lipids accumulated in unesterified forms within microglial membrane (lipid rafts) activate inflammatory responses through TLR and NF-kB signaling pathway [110]. Alongside, higher concentration of free cholesterol in the lysosomes ultimately form cholesterol crystals, inductors of lysosomal damage, and activators of NLRP3 inflammasome that consequently produce and release IL-1β and IL-18 inflammatory cytokines, as described in other inflammatory diseases (e.g., atherosclerosis) [111]. Concordantly, a distinct microglial subpopulation, the lipid-droplet accumulating microglia (LDAM), with dysfunctional intracellular lipid deposits was recently identified in the aging brain and similarly presented a unique transcriptional signature with increased production of pro-inflammatory cytokines, elevated levels of reactive oxygen species (ROS), and phagocytic deficiencies which ultimately reinforces previous research associating lipid disturbances with microglial pro-inflammatory shift [112]. In MS, the microRNA analysis profile from demyelinated lesions revealed the upregulation of microRNAs 34a, 155, and 326 inhibitors of myelin expressed “don’t eat me” protein CD47, the one that prevents macrophage activity through its interaction with the microglial receptor SIRPα [113,114]. Dysregulation of microRNA profile, upregulation of myelin microglial receptors within demyelinated areas together with the enormous amount of degraded myelin extracellularly accumulated over chronic demyelination can facilitate excessive and continuous myelin uptake with consequent formation of dysfunctional foamy microglia [113,115,116,117]. In line with this, not only myelin clearance improves disease recovery, but researchers have also tried to modulate cholesterol metabolism, accumulation, and intracellular transportation in order to prevent lipid-mediated harmful microglia response.

To begin with, studies show that heterodimeric transcription factor LXR-RXR is involved in the regulation of cholesterol balance and metabolism in the brain through transcriptional upregulation of its target gene ABCA1 [117]. Cantuti-Castelvetri et al. using a lysolecithin-induced demyelinating model, observed that old mice failed to resolve the inflammatory response and had limited restoration and remyelination capacity due to their decreased ability to phagocytose myelin debris alongside their abnormal accumulation of cholesterol crystals. However, when treated with the GW3965 molecule, an agonist of the transcription factor LXR, the authors observed ameliorated lipid efflux, enhanced lesion repair, and decreased microglial activation in old mice, which strongly supports the finding of new molecules targeting myelin degradation pathways to reduce cholesterol/lipid accumulation and lysosomal dysfunction for MS therapeutics [118]. Similar to GW3965, other LXR agonists, T0901317 and N,N-dimethyl-3β-hydroxy-cholenamide (DMHCA), that also increased ABCA1 expression in foamy macrophages and even prevented plaque formation in apolipoprotein E-deficient mice (animals with unbalanced cholesterol metabolism in macrophages prone to the formation of inflammatory atherosclerotic plaques) [119,120,121], may have a beneficial role. Additionally, the PPARγ signaling pathway modulates the activation of LXRα, so that its regulation can indirectly interfere with cholesterol efflux pathways [122]. In fact, studies from Chinetti and colleagues revealed that the treatment of both primary human monocyte-derived macrophages and macrophage-derived foam cells with PPAR-α/γ activators-Wy14643 or rosiglitazone and troglitazone-upregulated ABCA1 mRNA levels again interfere with the cholesterol efflux. In the same study, treatment with such PPAR agonists similarly decreased lipid accumulation in monocyte-derived macrophages previously treated with AcLDL, to facilitate cholesteryl-ester accumulation [123].

Other studies approached the histone deacetylase (HDAC) inhibitors for their considerable role in the regulation of NPC1 and 2 gene expression. In vitro studies using NPC1 mutant fibroblasts (with accumulated unesterified cholesterol) treated with LBH589 and Vorinostat, both HDAC inhibitors, increased the expression of NPC1 gene and restored cholesterol homeostasis [124,125]. Even more interesting, fingolimod (FTY720), approved for clinical treatment of MS, is also a HDAC inhibitor. From the Newton et al. study, FTY720-treated mice showed augmented expression of NPC1 and 2 in the hippocampus, cerebellum, and liver, which correlated with decreased liver cholesterol levels after treatment. In the same study, human NPC1 mutant fibroblasts incubated with FTY720 not only increased mRNA expression of NPC membrane proteins but markedly upregulated ABCAA1 membrane lipid transporter, which in turn decreased cholesterol accumulation within mutant cells [126]. Concordantly, FTY720 reduced total cholesterol content in induced-foamy human primary macrophages [127]. Such dual role of FTY720 as immunomodulatory and mediator of intracellular cholesterol trafficking likely undergird its observed beneficial role over remyelination after ex vivo demyelination [128] and microglial activation in MS focal inflammatory lesions [129].

Some of the discussed molecules (resumed in Table 1) require further testing using MS-disease models. However, based on their successful results in restoring lipid metabolism and imbalance, they present a good therapeutic opportunity alone or in a combined manner with phagocytose-enhancing drugs, to overcome remyelination impairment and, therefore, MS progressive disability.

#### 2.2.3. Phagocytosis and Cognitive Impairment

The above proposed phagocytose-mediated therapeutic approaches to counteract the MS disease burden can also raise new challenges. Whereas the process of myelin clearance contributes to a less harmful environment within demyelinated plaques, recent studies have strongly correlated microglial phagocytosis with cognitive impairment in MS, for what there is also the loss of treatment perspectives.

In addition to the well visible motor disability of MS patients, cognitive impairment is increasingly recognized as one of the primary features of MS (affecting around 40–60% of patients). This “invisible” disability that is associated with cortical lesions and atrophy over multiple brain regions may have an early onset, significantly affecting the patient’s social interactions, ability to work, and quality of life [130,131]. Studies from Di Filippo and colleagues associate inflammatory responses and its subsequent anomalous release of inflammatory mediators (as in MS) with disruption of glia-neuron crosstalk [131]. Cytokines, acting as neuromodulators, impair synaptic transmission, long term potentiation maintenance, and synaptic plasticity, thus interfering with cognitive functions [132]. Particularly, elevated expression and release of TNF-α were observed to interfere with spontaneous excitatory postsynaptic currents and excessive glutamatergic release [133], ultimately associated with increased neurotoxicity contextual learning-memory impairment in EAE-induced mice [134,135]. On the other hand, demyelination potentiates axonal vulnerability to inflammatory damage, thus naturally contributing to neuronal loss and impaired synaptic communications. Particularly vulnerable is the hippocampus, the central region responsible for the maintenance and retrieval of memory where demyelination can be detected on more than 60% of postmortem MS hippocampi [136,137]. Studies from Dutta et al. also demonstrate that loss of myelin within hippocampal areas were associated with reduced expression of neuronal proteins involved in synaptic integrity, axonal transport, and glutamate homeostasis. This led to a decrease in synaptic density that is in line with a previous imaging study, using magnetic resonance imaging (MRI), correlating altered hippocampal measure and atrophy with memory dysfunction in MS patients [138,139].

However, recent studies using high-resolution confocal microscopy also revealed an extensive synaptic loss in the MS cortex, independent from demyelination and axonal degeneration, strongly implying microglial involvement in synaptic loss and cognitive dysfunction [140]. Indeed, activated microglia were associated with disturbances in glutamate uptake inducing neuronal and synaptic damage in cortical lesions [141] and early memory impairment in EAE mice [142]. Moreover, as described earlier, microglial phagocytosis mediates synaptic refinement during brain development, a process that reactivates in aged-related and neurodegenerative disorders such as AD [143]. Likewise, the microglial-mediated synaptic loss may eventually be involved in MS-associated cognitive impairment. Not only there is an enhanced expression of complement components, namely C1q and C3, over neuroinflammation [144] but also the “don’t eat me” signaling protein CD47 that also protects synapses from excessive phagocytosis is found downregulated in chronic MS lesions [114,145]. Consistent with this finding, studies from Michailidou et al. found complement proteins colocalizing with synaptophysin, either in neuronal synapses or inside activated microglia, within both demyelinated and myelinated MS hippocampi [146]. Concordantly, C3-knockout in mice inhibited synapse removal and protected hippocampal neuronal loss during normal aging, accompanied by increased learning and spatial memory when compared to age-matched WT mice [147]. Further relevant, blockage of C3 also preserved synapse density, protected hippocampal synaptic degeneration, and improved cognitive functions on a mouse model of AD [148].

Despite these advantages, the association of microglial role with the MS-characteristic cognitive dysfunction carries future studies and analysis. However, the dual role of complement signaling pathway both in the opsonization of myelin debris and tagging of synapses raises challenging questions: (1) If, by modulating debris phagocytosis and promoting myelin recovery, we will in fact be potentiating a secondary loss of synapses and long-term cognitive impairment; (2) whether it will be possible to modulate phagocytosis independently of the complement system to increase myelin clearing while avoiding extreme synaptic pruning. Thus, the in-depth study of the regulation of synapses phagocytosis while the formation of MS lesions becomes demanding so that we can invest in possible safe therapies targeting myelin clearance without promoting additional neuronal dysfunction and synaptic loss.

## 3. Conclusions

For long, microglia have been the subject of much research in MS regarding their dynamic functions and dual inflammatory role over disease progression. Whereas as pro-inflammatory cells they contribute to disease burden, in an anti-inflammatory and regenerative state they are essential for disease recovery. Based on such unbalanced phenotypic dualities which underlie MS pathology, successful anti-inflammatory therapies have been focused on targeting the microglial immune role, while favoring at the same time, the switch towards the most beneficial phenotype preserving their regenerative functions. On the contrary, therapeutics regarding the microglial main function of phagocytosis have been less explored. In this review, we summarize the contribution of phagocytic process to early development, however focusing on its functional consequences over MS disease progression, not only in myelin removal from demyelinated lesions, but also as a modulator of microglial inflammatory responses. Therefore, the microglial phagocytic dysfunction results in severe motor and cognitive MS-associated symptoms so that we highlight newly phagocytic-directed approaches, which we discuss that in combination with anti-inflammatory molecules may achieve promising results regarding MS therapeutics. Importantly, the challenge will be to find a balance between complement-mediated myelin clearance and synaptic removal in order to find successful phagocytosis-targeted drugs and their optimization towards clinical applications to ameliorate disease pathogenesis or even promote the full patient’s recovery.

## Figures and Tables

**Figure 1 ijms-21-05960-f001:**
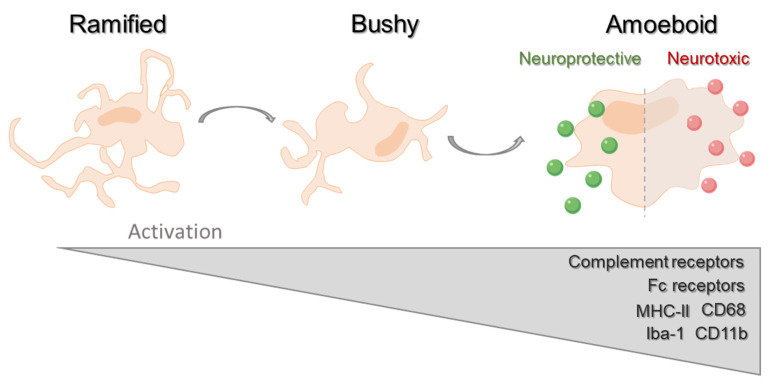
Schematic representation of microglial activation. Following any insult or alteration in brain homeostasis microglia become activated. Morphologically, cells partially retract their processes evolving from a ramified state, to an intermediate one or “bushy” and finally converting into a complete rod and amoeboid shape or fully activated state. In parallel, activated microglia overexpress specific characteristic markers such as ionized calcium-binding adaptor molecule-1 (Iba-1), major histocompatibility complex antigen class II (MHC-II), and microglial phagocytic markers such as CD11b, complement and Fc receptors I–III, and CD68. Finally, these activated cells can evolve into functionally distinct phenotypes either becoming neurotoxic or neuroprotective-reactive states associated with the release of both pro- and anti-inflammatory mediators and cytokines, respectively.

**Figure 2 ijms-21-05960-f002:**
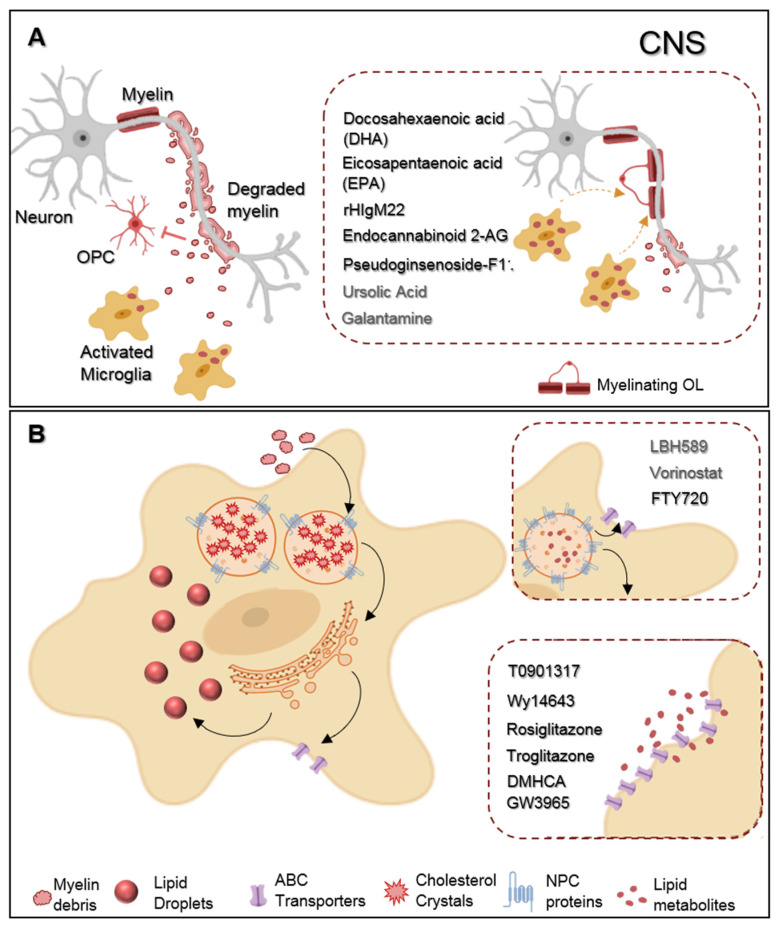
Representative scheme of microglia myelin phagocytosis modulating drugs. (**A**) During multiple sclerosis (MS) progression myelin is degraded into myelin-toxic debris within demyelinated plaques in the central nervous system (CNS). Giving debris toxicity towards oligodendrocyte (OL) precursor cells (OPCs) inhibiting their differentiation into full differentiated myelinating OLs, myelin debris must be cleared away by microglia. Regarding therapeutics, docosahexaenoic acid (DHA) and eicosapentaenoic acid (EPA) increase myelin uptake by microglia. Endocannabinoid 2-AG favors both myelin microglial clearance and OPC differentiation. Pseudoginsenoside-F11 accelerates CR3-dependent myelin phagocytosis while rHIgM22 binds to myelin debris and also facilitates their entrance towards microglia. Galantamine increases microglial uptake of Aβ aggregates and ursolic acid can interfere in the upstream regulation of CD36 expression, so that they may likely have an effect on myelin phagocytosis. (**B**) Demyelinated lesions are associated with foamy phagocytes presenting a pro-inflammatory profile as a result of over-internalization of cholesterol-rich myelin debris. Intracellularly, excessive cholesterol dysregulates lipid metabolism and efflux pathways and accumulates in the lysosomes forming cholesterol crystals or can be stored into lipid droplets. Proposed therapeutics are based on the functionalization of lipid metabolism and associated pathways. LBH589 and Vorinostat increase the expression of NPC proteins in fibroblasts, facilitating cholesterol release from lysosomes, for what we envision a possible effect on microglia as well. FTY720 treatment lead to the overexpression of both NPC and ABC transporters also promoting lipid efflux from human primary macrophages. DMHCA and GW3965 directly or Wy14643, Rosiglitazone, and Troglitazone, indirectly, activate the LXR/RXR heterodimeric transcription factor which upregulates its transcriptional targets ABC transporters then again facilitates cholesterol/lipid exit from macrophages.

**Table 1 ijms-21-05960-t001:** List of possible drugs to modulate microglia towards a regenerative phenotype in multiple sclerosis.

**Modulators of Microglial Phagocytic Phenotype**		
**Drug**	**Mechanism**	**Reference**
Docosahexaenoic acid and Eicosapentaenoic acid	Enhance myelin phagocytosis by microglia	[87]
Endocannabinoid 2-AG	Upregulates the expression levels of phagocytosis associated genes and promotes microglial myelin uptake	[92]
Pseudoginsenoside-F11	Accelerates CR3-dependent myelin phagocytosis by microglial cells	[93]
rHIgM22	Binds to myelin debris and facilitates their internalization by microglia	[88]
Ursolic acid ***	Agonist of PPARγ signaling, which upregulates the expression of CD36 receptor, involved in the internalization of Aβ and lipids	[97]
Galantamine ***	Favors microglial Aβ internalization	[99]
**Modulators of Microglial Lipid Metabolism**		
**Drug**	**Mechanism**	**Reference**
LBH589 ^#^ and Vorinostat ^#^	Both HDAC inhibitors that increase NPC expression in fibroblasts, promoting the release of cholesterol from lysosomes	[124,125]
FTY720	HDAC inhibitor that also promotes lipid efflux from human primary macrophage through the overexpression of both NPC proteins and ABC transporters	[126,127]
DMHCA, GW3965, Wy14643, Rosiglitazone and Troglitazone	Activators of LXR/RXR heterodimeric transcription factor which upregulates ABC transporters thus also promoting cholesterol/lipid exit from macrophages	[118,119,120,121,122,123]

*** Both Ursolic acid and Galantamine affect cell phagocytosis, for what we highlight them as good candidates to modulate myelin phagocytosis. ^#^ The effect of LBH589 and Vorinostat on the expression of NPC proteins in microglia is unknown. Further studies are needed to evaluate the potential of these molecules on microglial lipid metabolism. CR3: Complement receptor 3; PPAR: Peroxisome proliferator-activated receptor; Aβ: Amyloid-β; HDAC: Histone deacetylase; NPC: Niemann-Pick type disease C; ABC: ATP-binding cassette; LXR: liver X receptor; RXR: Retinoid X receptor.

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
