# Peer review of "Microglial Phagocytosis—Rational but Challenging Therapeutic Target in Multiple Sclerosis"

_ijms, 2020, doi:10.3390/ijms21175960_

Round 1
Reviewer 1 Report
This is an interesting review article on the role of microglial phagocytosis in MS and potential new avenues towards pro-regenerative treatments. I indeed learned a lot from this overview and have only minor comments:
Please rather refrain from dividing microglia into M1 and M2 classes, this nomenclature is outdated and does not really reflect the complexity of microglial phenotypes as nicely revealed in a number of recent (transcriptome) studies.
The manuscript would probably benefit if the drugs related to inflammatory phenotypes vs. phagocytitic phenotypes could be additionally presented in a table.
Figure 1 (within grey triangle): exchange Cd11B to CD11b (as in text).
Author Response
We thank Reviewer for this valuable comments. As suggested we removed the M1 and M2 nomenclature and maintained only the more neurotoxic and/or neuroprotective classification in the text (as highlighted in page 2- line 87 and page 3- line 104), and figure 1 (in page 3), as well as in Figure legend (line 106).
In addition, we also added a table with the overall information about all the described drugs in page 10, as outlines in the text in page 9- line 382.
Finally, CD11b in Figure 1 was modified accordingly, at page 3.
Reviewer 2 Report
This is a clearly written and well-organized review. It summarizes the latest findings on the beneficial role of microglia on remyelination by clearing the myelin debris. It also emphasizes the need of deeper studies and a better understanding of this microglial role to promote new therapeutic approaches for multiple sclerosis, by promoting remyelination in addition to the inflammatory response control. This review is relevant and important for the audience to read and to take into consideration for their future research. However, some minor points should be addressed before publication.
Minor points:
In figure 1 some headings (“Ramified, BUSHY, Amoeboid”) are written in capital letters while the others are in lowercase. This should be homogeneous
In page 9 line 350, the reference {Zhao, 2010 #114} should be properly formatted.
Author Response
We thank to Reviewer for the valuable comments. As suggested, we changed the headings in Figure 1 to “Ramified, Bushy, Amoeboid” and properly formatted the above reference to [117], as highlighted in grey (page 9 – line 349).
Reviewer 3 Report
Since I am approving accepting the manuscript in its presence form I do not have specific comments to the authors expecting a reply. Nevertheless, the authors must provide the complete reference No. 8 ("Compston A, C.A. Multiple Sclerosis 2008")
Author Response
We also thank to Reviewer for the acceptance and useful comment. In accordance we have modified the reference, as highlighted, page 13 – line 490.